# T4-like Bacteriophages Isolated from Pig Stools Infect *Yersinia pseudotuberculosis* and *Yersinia pestis* Using LPS and OmpF as Receptors

**DOI:** 10.3390/v13020296

**Published:** 2021-02-13

**Authors:** Mabruka Salem, Maria I. Pajunen, Jin Woo Jun, Mikael Skurnik

**Affiliations:** 1Department of Bacteriology and Immunology, Medicum, Human Microbiome Research Program, Faculty of Medicine, University of Helsinki, 00290 Helsinki, Finland; mabruka.salem@helsinki.fi (M.S.); maria.pajunen@helsinki.fi (M.I.P.); 2Department of Microbiology, Faculty of Medicine, University of Benghazi, Benghazi 16063, Libya; 3Department of Aquaculture, Korea National College of Agriculture and Fisheries, Jeonju 54874, Korea; advancewoo@snu.ac.kr; 4Division of Clinical Microbiology, Helsinki University Hospital, HUSLAB, 00290 Helsinki, Finland

**Keywords:** *Y. pseudotuberculosis*, bacteriophage, receptor, *Myoviridae*, lipopolysaccharide, tail fiber, Gp38

## Abstract

The *Yersinia* bacteriophages fPS-2, fPS-65, and fPS-90, isolated from pig stools, have long contractile tails and elongated heads, and they belong to genus Tequatroviruses in the order *Caudovirales*. The phages exhibited relatively wide host ranges among *Yersinia pseudotuberculosis* and related species. One-step growth curve experiments revealed that the phages have latent periods of 50–80 min with burst sizes of 44–65 virions per infected cell. The phage genomes consist of circularly permuted dsDNA of 169,060, 167,058, and 167,132 bp in size, respectively, with a G + C content 35.3%. The number of predicted genes range from 267 to 271. The phage genomes are 84–92% identical to each other and ca 85% identical to phage T4. The phage receptors were identified by whole genome sequencing of spontaneous phage-resistant mutants. The phage-resistant strains had mutations in the *ompF, galU, hldD,* or *hldE* genes. OmpF is a porin, and the other genes encode lipopolysaccharide (LPS) biosynthetic enzymes. The *ompF*, *galU*, and *hldE* mutants were successfully complemented *in trans* with respective wild-type genes. The host recognition was assigned to long tail fiber tip protein Gp38, analogous to that of T-even phages such as *Salmonella* phage S16, specifically to the distal β-helices connecting loops.

## 1. Introduction

*Yersinia pseudotuberculosis* is a Gram-negative, zoonotic pathogen that occurs primarily in the northern hemisphere [1], and it causes yersiniosis which is manifested by fever, acute abdominal pain due to mesenteric lymphadenitis, and can be complicated with reactive arthritis and erythema nodosum [1,2,3]. Although infection with *Y. pseudotuberculosis* is usually self-limited, it can be accompanied with bacteremia, gastrointestinal bleeding or acute kidney injury [4], and in some cases can be misdiagnosed with acute appendicitis leading to unnecessary appendectomy [1,2]. Humans can get the infection by direct contact to the infected animals or via consumption of contaminated water or vegetables such as iceberg lettuce and carrots. Periodic outbreaks of *Y. pseudotuberculosis* are not uncommon, and they are more frequent at schools and day-care centers, which is considered as a major health problem as huge number of students and children can be infected [1]. In Finland, *Y. pseudotuberculosis* infection is a notifiable disease [3], and most of the outbreaks are caused by fresh produce [1]. As a psychrophilic organism, *Y. pseudotuberculosis* can grow at 4 °C, which is regarded as a potential food safety risk [5]. *Y. pseudotuberculosis* is composed of 15 serotypes, O:1 to O:15, of which serotypes O:1 and O:2 are further divided into subtypes a, b, and c, while O:4 and O:5 into a and b, respectively [6].

Bacteriophages (phages) are viruses that infect bacteria, and can be used instead of antibiotics to control bacterial infections. Most phages carry a double-stranded DNA genome, that is packaged into a protein capsid, the head, to which is attached a tail that mediates the interaction between the host bacteria and the phage. Phages start the host infection through interaction between the receptor-binding protein of the phage usually located at the tip of the tail or the tail fibers. The receptor-binding protein binds specifically to a typical bacterial surface structure such as polysaccharide or protein. In Gram-negative bacteria these are typically lipopolysaccharides (LPS) or porins, and, for example, these both are used as receptors of two closely related myoviridae, phages ϕR1-RT and TG1, that infect *Yersinia enterocolitica* [7].

Very few bacteriophages that infect *Y. pseudotuberculosis* have been described to date. In this work, we studied three phages infecting *Y. pseudotuberculosis* serotype O:1a which were isolated from pig stools collected from three different pig farms in Finland [8]. The aim of this study was to characterize these phages so that they could be used in future for detection and control of yersiniosis. To our current knowledge, this is the first comprehensive and detailed study on phages infecting *Y. pseudotuberculosis* O:1a bacteria. Generally, the studied phages, named fPS-2, fPS-65, and fPS-90, were characterized, sequenced, and annotated.

## 2. Materials and Methods

### 2.1. Bacteria, Plasmids, Phages, and Media

The phages were isolated from pig stool samples collected from different Finnish pig farms as described previously [8]. The phages were propagated on Lysogeny broth (LB), Lysogeny agar (LA) plates (LB with 1.5% agar), and soft agar (LB with 0.4% agar). In certain situations, Yersinia-selective agar (=CIN agar, Cefsulodin-Irgasan-Novobiocin agar) and Tryptic Soy Broth (TSB) were used. The bacterial strains and plasmids used in this work are listed in Appendix A, respectively. When appropriate, antibiotics were added to the growth media at the following concentrations: chloramphenicol (Clm), 20 µg/mL; tetracyclin (Tet), 12.5 µg/mL, and kanamycin (Km), 100 µg/mL in agar plates and 20 µg/mL in broth. When necessary, diaminopimelic acid (DAP) was supplemented to a final concentration of 0.3 mM.

### 2.2. Electron Microscopy

The phage particles were concentrated by centrifugation at 16,000× *g* for 90 min at 4 °C using an Eppendorf centrifuge (5415R, rotor model 3328, Enfield, NJ, USA). A carbon-coated copper grid was incubated with a drop of the phage preparation for one minute and then stained with 1% uranyl acetate (pH about 4.2) for 30 s. The micrographs were then captured using a transmission electron microscope (JEOL JEM-1400, Tokyo, Japan, 80 kV) with an Olympus Morada CCD-camera (Olympus, Tokyo, Japan), operating at iTEM software (EMSIS GmbH, Muenster, Germany).

### 2.3. Phage DNA Isolation

The phage genomic DNAs were isolated as described in a previous study [8]. Briefly, to a 500 µL of phage suspension, 1 µL DNase I (10 mg/mL) and 3 µL RNase A (10 mg/mL) were added and incubated at 37 °C for 1 hr to degrade the bacterial nucleic acids. Then 12 µL of 0.5 M EDTA, 1 µL of proteinase K (20 mg/mL), and 20 µL of 10% SDS were added to the mixture and incubated at 56 °C for 1 hr with shaking. The suspension was cooled down to room temperature (RT, 22 °C) before adding 1 volume of phenol. The mixture was vortexed and then centrifuged at RT for 5 min. The water phase was extracted with 1 volume phenol:chloroform:isoamyl alcohol (25:24:1) and then re-extracted with 1 volume of chloroform: isoamyl alcohol (24:1). The phage DNA was precipitated with 60 µL of 3 M sodium acetate pH 7.0 and 1.2 mL of cold absolute ethanol (EtOH). The DNA pellet was washed with 1 mL of 70% EtOH, then the dried pellet was dissolved in 50 µL of distilled water and stored at 4 °C. DNA concentration and quality were determined by spectrophotometer (NanoDrop spectrophotometer ND-1000, Wilmington, NC, USA).

### 2.4. Phage Genomes Sequencing and Bioinformatics

The phage genomes were sequenced using the Illumina Miseq platform at FIMM (https://www.fimm.fi/en/services/technology-centre/sequencing) (accessed on 15 October 2015), resulting in >200-fold coverage. The raw sequence data were assembled using the de novo assembly A5-miseq pipeline [9]. The putative genes of each genome were identified by RAST [10] and Glimmer [11], and confirmed by manual inspection using the Artemis tool [12]. BLASTP [13] was used to compare the gene products of fPS-65, fPS-90 and phage T4 to the corresponding gene products of fPS-2. The fPS-phage genome sequences along with that of T4 were aligned using clustal format alignment by MAFFT [14]. The search for tRNAs was performed using tRNAScan-SE [15,16]. Phylogenetic trees of the three phages were constructed using VICTOR [17]. Putative phage promoter sequences were identified by searching for homologous sequences to the conserved promoter sequences of T4 phages within the 100 upstream region of each predicted gene. The putative phage terminator sequences were identified using Arnold’s tool (http://rssf.i2bc.paris-saclay.fr/) (accessed on 11 April 2019), with further manual verification.

### 2.5. Host Range and Efficiency of Plating (EOP)

The ability of the three phages to infect representative strains of order *Enterobacteriales* and selected LPS mutant strains of *Y. pseudotuberculosis* and *Yersinia pestis* (Appendix A) was tested using the spot method [18]. Briefly, 5 µL aliquots of serially 1:10-diluted phage stock were spotted onto bacterial lawns prepared with soft agar overlay. After overnight incubation at RT, the plates were inspected for the presence of lysis zones. To determine the EOP values of the phages for different bacterial strains, 40 µL of the last phage dilution that showed discrete plaques was mixed with 90 µL of the bacterial culture (OD_600_ ~1.0) and 3 mL of soft agar, and poured on LA plate to calculate the exact number of plaques. To calculate the EOP value, the phage titer on the test bacteria was divided by the phage titer on the original host bacteria [19].

### 2.6. One-Step Growth Curves

One-step growth curve experiments were performed to determine the latent time and the burst sizes of the phages according to the previously published method [18]. Briefly, the phages were added to a mid-exponential phase culture of *Y. pseudotuberculosis* strain PB1 host bacteria at multiplicity of infection (MOI) of approximately 10^−5^ and allowed to adsorb for 10 min at RT. The mixture was then centrifuged to remove the non-adsorbed phages, and the pellet was resuspended in fresh TSB; from which subsequent 1:10, 1:100, and 1:1000 dilutions were prepared immediately. From these dilutions, samples were collected every 10 min (up to 2 h), then immediately diluted and plated for phage titer quantifying using the double-layer agar method and *Y. pseudotuberculosis* strain PB1 as indicator bacteria. The experiments were performed at least twice with two replicates.

### 2.7. Phage Growth Curves Using the Bioscreen Incubator

The experiment was performed as previously described [20] with some modifications. Briefly, overnight culture of *Y. pseudotuberculosis* serotype O:1a strain PB1 was diluted 1:10 in fresh LB medium to an initial OD_600_ value of 0.2; from which 180 µL aliquots were distributed into honeycomb plate wells (Growth Curves Ab Ltd., Helsinki, Finland) to which the phages fPS-2, fPS-65, and fPS-90 were added in 20 µL aliquots resulting in approximate MOIs of 0.001 for each phage. Positive controls contained 180 µL of the bacterial cultures and 20 µL of medium, while the negative controls were prepared by mixing 180 µL of medium with 20 µL of phage stock. The experiment was carried out at RT for about 65 h using the Bioscreen C incubator (Growth Curves Ab Ltd., Helsinki, Finland) with continuous shaking. The optical density OD_600_ values were measured at selected time intervals. For each condition, the experiment was carried out in five parallel wells.

### 2.8. Heat-Stability of Phage Receptor

Overnight culture of *Y. pseudotuberculosis* O:1a strain PB1 was diluted 1:5 in fresh LB medium, and 1 mL of this bacterial suspension was boiled at 100 °C for 10 min, immediately cooled on ice, and then serially diluted (1:2, 1:4, 1:8, and 1:16) in LB medium. Similarly treated non-boiled live bacteria were used as positive controls. A negative control was prepared by replacing the bacterial culture with 100 µL growth medium alone. To each dilution (100 µL) and the control, the same amount of phage fPS-65 was added (about 140 PFUs). After incubation for 15 min with gentle shaking, the mixtures were centrifuged for 3 min to remove the adsorbed phage particles. The numbers of free phages in the supernatants were determined to reveal whether heat-treatment had destroyed the phage receptor. The supernatant samples were mixed with the *Y. pseudotuberculosis* PB1 bacteria and then added to soft agar tube and spread onto LA plates. The plates were incubated at RT and the number of plaques was counted the next day. The PFU value of the negative control was set to 100% and the other values were calculated related to that.

### 2.9. Isolation of Spontaneous Phage-Resistant Mutants and Analysis of Their Genomes

Twenty µL aliquots of the phages fPS-2 and fPS-90 (about 10^7^ PFU) were pipetted on fresh *Y. pseudotuberculosis* PB1 lawns on LA plates. Spontaneous phage-resistant mutants growing as colonies within the lysis zones were isolated after incubation of the plates at RT for at least three days. The selected colonies were re-streaked individually on fresh LA plates at least twice to guarantee pure cultures. The phage-sensitivity of the recovered strains was re-tested using the spot droplet method to ensure their resistance phenotypes. The confirmed phage-resistant mutants were stored for further experiments. With phage fPS-65, however, this plate isolation approach was not successful; all the isolated colonies turned out to be still sensitive to the phage, indicating that the spontaneous mutants are rare. To enrich the mutants, we co-cultured the bacteria and the phage in liquid culture simulating the procedure of the growth-curve experiments carried out with the Bioscreen (see Section 2.7). Thus, 200 µL of fPS-65 (~ 10^7^ PFU) was mixed with 1.8 mL of the 1:10 diluted overnight culture of the *Y. pseudotuberculosis* O:1a strain PB1 bacteria (~10^9^ CFU) and incubated with aeration at RT for 48 h. The bacteria were pelleted by centrifugation, washed, and resuspended into 2 mL of fresh LB from which 50 µL drops were spread on LA plates. After incubation for six days at RT, twelve colonies were picked up and cultured on fresh *Yersinia*-selective CIN agar plates to prevent the growth of bacterial contaminants. The re-streaking was repeated at least twice to ensure pure cultures, and their resistant phenotype was tested as above.

The genomic DNA of the phage-resistant bacterial mutants was isolated and purified using JetFlex Genomic DNA Purification Kit (Thermo Fisher Scientific,Waltham, MA, USA) following the manufacturer’s instructions. The obtained DNA was subjected to sequencing at Novogene Europe (Cambridge, UK) using Illumina HiSeq with 150-bp paired-end reads. The obtained sequence reads of the phage-resistant strains were aligned to the genome sequence of the parental *Y. pseudotuberculosis* O:1a strain PB1 (Acc no NC_010634) using the “Map to reference” -tool of the Geneious 10.2.6 (www.geneious.com) (accessed on 20 September 2020). The resulting alignments of the reads were then manually inspected to identify the differences.

### 2.10. Construction of the ompF Complementation Plasmid pTM100-ompF

The wild type *ompF* gene of *Y. pseudotuberculosis* O:1a strain PB1 including its upstream promoter region was amplified with Phusion DNA polymerase using primers PB1ompF-F (5’-cgcggatccagctctgctggcttttat-3’) and PB1ompF-R (5’-cgcggatccagaggcttccatggcttag-3’). The obtained 1592 bp fragment was digested with BamHI and ligated into BamHI digested, alkaline phosphatase-treated pTM100 plasmid. The ligation mixture was electroporated into *E. coli* DH10B cells. The proper insertion of the *ompF* gene in the resulting plasmid pTM100-ompF was verified by restriction digestions and by sequencing with pTM100 specific primer ptmcla-f (5′-caaatgtagcacctgaagtc-3′).

### 2.11. Conjugation and Complementation

Triparental conjugation was performed as previously described [21]. Briefly, the plasmid-carrying *E. coli* donor strains were grown without shaking overnight at 37 °C in 10 mL of LB broth supplemented with appropriate antibiotic, while the helper strain *E. coli* HB101/pRK2103 was grown in 10 mL of LB broth supplemented with Km overnight at 37 °C with shaking. The recipient *Y. pseudotuberculosis* phage-resistant mutant strains were grown with shaking overnight at RT in 10 mL LB broth. The overnight bacterial cultures were centrifuged (1500× *g*, 10 min), the pellets washed twice with PBS (phosphate-buffered saline) to remove the remaining antibiotics, and then resuspended in 1 mL of PBS to OD_600_ ~1.0. The donor, recipient, and helper bacteria were mixed 1:1:1 and 50-µL drops of the obtained mating mixtures were spread on LA plates. After 24 h of incubation at RT, the mating mixtures were collected and suspended in 1 mL PBS, from which serial dilutions were plated on CIN agar plates supplemented with appropriate antibiotic to select for the successful *Y. pseudotuberculosis* transconjugants. Colonies growing on the CIN agar plates were picked up, and pure cultures were prepared on LA plates supplemented with appropriate antibiotic. To complement the fPS-65-resistant mutants with the *hldE* gene carrying plasmid pTM100-hldE, the same protocol was followed but without the use of the helper strain. The growth media for the donor strain *E. coli* ω7249/pTM100-hldE were supplemented with DAP in addition to the selective antibiotic. The obtained *ompF*-, *galU*- and *hldE*- complemented strains were stored for further use.

### 2.12. Accession Numbers

The whole genome sequences of fPS-2, fPS-65, and fPS-90 were deposited in European Nucleotide Archive under the accession numbers: LR215722, LR215724, and LR215723, respectively.

## 3. Results and Discussion

### 3.1. Electron Microscopy

When visualized under the transmission electron microscope (Figure 1), the phages were having long contractile tails with elongated heads suggesting that they belong to the genus Tequatroviruses, *Myoviridae* family under the *Caudovirales* order. Overall the dimensions of the phage particles are close to that of phage T4 with head diameter of 85 nm, head length of 115 nm, and tail length of ca 100 nm [22]. The latter reflects perfectly the identical sizes of 590 amino acid residues for the tape measure proteins of the fPS-phages and that of T4.

### 3.2. Annotation and Analysis of the Phage Genomes

The complete genomes of the three phages were obtained after performing the de novo assembly and manual refinement of the raw sequenced data. The sizes of the linear double stranded genomes were 169,060, 167,058, and 167,132 bp for fPS-2, fPS-65, and fPS-90, respectively; with an average G + C content of 35.3%, which is close to the G + C content of T4 phages (34.5%) [23]. The phage genomes are 84–92% identical to each other and ca 85% identical to phage T4 (Figure 2). Although we did not experimentally address it here, the fPS-phages very likely contain linear, circularly permuted DNA, similar to phage T4 [23]. For phage fPS-2, the coding regions of the whole genome was 93.7% with 267 predicted genes; 223 of which lie on the reverse strand and 44 on the forward strand. For fPS-65, 268 of predicted genes and 93.7% coding percentage were predicted with 227 genes on the reverse strand and 41 on the forward strand. The total number of predicted genes for phage fPS-90 was 271 of which 230 on the reverse strand and 41 on the forward strand. A cluster of eight tRNA genes was detected for fPS-2 and fPS-90, and 11 tRNA genes was predicted for fPS-65. It is common to find clustered tRNA-related genes in phages with large genomes, as the *Myoviridae* phages, and it is believed that they facilitate the translation process [24]. The comparative analysis of the gene products of the phages to each other and to T4 phage (accession no. AF158101) revealed high similarity at the level of amino acid sequences and their arrangement (Figure 2 and Appendix A). Phages usually have a sequential arrangement of different modules, and genes in each module are arranged very closely to form a gene cluster of specific function [25]. Accordingly, we could identify genes encoding products belonging to DNA replication and nucleotide biosynthesis, virion morphogenesis, DNA packaging, and host lysis organized very close to that of the T4 genome (Figure 2). The absence of lysogenic genes in the phage genomes supports the conclusion of the lytic nature of these phages.

### 3.3. Phylogenetic Analysis

A phylogenetic tree (Figure 3) for the whole phage genomes of 15 phages was constructed using the VICTOR Virus Classification and Tree Building Online Resource [17] using the Genome-BLAST Distance Phylogeny (GBDP F0) method [27] under settings recommended for prokaryotic viruses [17]. Eight of the selected phage genomes (*Escherichia* virus T4 “AF158101”, *Escherichia* virus ECML-134 “NC_025449”, *Escherichia* virus slur04 “NC_042130”, *Enterobacteria* phage RB3 “NC_025419”, *Shigella* virus Shfl2 “NC_015457”, *Shigella* virus pss1 “NC_025829”, *Yersinia* virus D1 “NC_027353”, *Yersinia* virus PST “NC_027404”) represent the *Tequatrovirus* genus. The other seven phage genomes, *Escherichia* virus BP7 (NC_019500), *Enterobacter* phage PG7 (NC_023561), *Klebsiella* virus JD18 (NC_028686), *Salmonella* virus S16 (NC_020416), *Shigella* virus SP18 (NC_014595), *Citrobacter* virus CF1 (NC_042067) and *Vibrio* phage ValKK3 (NC_028829) represent different genera of the *Myoviridae* family.

The phylogeny software revealed four clusters, where fPS- phages along with the phages belonging to the genus *Tequatrovirus* were grouped within the same cluster while the other phages from the other genera were grouped in a separate cluster. Unexpectedly, fPS-65 and fPS-90 were grouped within the same sub-cluster while fPS-2 in a more distant sub-cluster although fPS-2 is more similar to fPS-90 than fPS-65.

### 3.4. Host Range and EOP

Altogether, 94 bacterial strains representing four different *Yersinia* species and eight other bacterial genera were used to study the host range of the phages (Appendix A). In addition, LPS mutants of *Y. pseudotuberculosis* and *Y. pestis* were used in order to identify the phage receptor. Phage fPS-65 showed the broadest host range among the three phages, as it had the ability to infect 33 different *Y. pseudotuberculosis* strains, while fPS-2 and fPS-90 infected only 21 and 18 strains, respectively, with EOPs ranging from 10^−1^ to 10^−7^. In addition, fPS-65 could lyse the O:2a serotype strain Hatada A as efficiently as its original *Y. pseudotuberculosis* O:1a host strain PB1. Furthermore, fPS-65 was also able to infect all the nine tested *Y. pestis* strains including the deep rough mutants, though less efficiently than the *Y. pseudotuberculosis* strain PB1. Both fPS-2 and fPS-90, in contrast to fPS-65, could infect *Y. similis* O:12 strain N916Ysi. Among the strains representing other bacterial genera, only the *E. coli* O:1 K:1 strain IHE11002 was sensitive to all the tested fPS-phages. In general, these results indicated that the phages have a relatively broad host range as they could infect many bacterial strains within the same bacterial species in addition to a strain from another genus (*E. coli*) [28]. However, such results did not give us a clear hint about the phage receptors, especially for fPS-65 which was able to infect many LPS mutant strains of *Y. pseudotuberculosis* and *Y. pestis* with different levels of truncations at the LPS structure.

In general, host range determination has been studied for many T4-like phages; for example, Vibriophage KVP40 has a broad host range, infecting some pathogenic and non-pathogenic *Vibrio* species [29]. Additionally, the coliphages AR1 and LG1 have a wide host range as they are able to lyse many serogroups of *E. coli* and other members of enterobacteria [30]. On the other hand, some studies have reported T4 phages with narrow host ranges, as vB_Kpn_F48, the *K. pneumoniae* phage, which has a lytic activity specific to *K. pneumoniae* strains of ST101 [31].

### 3.5. Growth Characteristics

The one-step growth study revealed that fPS-2 and fPS-65 had similar latent period of 50–55 min, with different burst sizes of about 65 and 44 virions per infected bacterium, respectively (Figure 4). Phage fPS-90 showed a bit different growth kinetics of 75–80 min latent period and a burst size of about 60 virions per infected cell (Figure 4).

Of note, there is a remarkable diversity of growth kinetics among T4-like phages; i.e., the latency periods and burst sizes vary from phage to other. For example, phage vB_Kpn_F48 has a short latent period of less than 10 min with a burst size of 72 phage particle per infected bacterium [31]. A similar short latent period was computed for the *Enterobacter aerogenes* phage vB_EaeM_φEap-3, with a higher burst size of 109 PFU per infected cell [32]. On the other hand the coliphages, LG1 and AR1, had different growth kinetics as the latent periods were 52 and 40 min, and the burst sizes 177 and 38 PFU per infected cell, respectively [30].

### 3.6. Identification of the Phage Receptors

In order to identify the receptors on bacterial surface to which the phages adsorb, we isolated 15 spontaneous phage-resistant mutants of *Y. pseudotuberculosis* O:1a strain PB1. Three mutants, resistant to fPS-2 were named M1-fps2-wt, M2-fps2-wt, and M3-fps2-wt. Nine mutants resistant to fPS-90 were named M1-fps90-wt, M3-fps90-wt, M4-fps90-wt, M5-fps90-wt, M6-fps90-wt, M7-fps90-wt, M8-fps90-wt, M9-fps90-wt, and M10-fps90-wt. The three mutants resistant to fPS-65 were named M1-fps65-wt, M2-fps65-wt, and M3-fps65-wt (Table 1). To confirm the resistance phenotype of the mutant strains, we tested their sensitivity also to the other fPS-phages. Generally, all the fPS-90-resistant strains were resistant to both fPS-90 and fPS-2 with the exception of M4-fps90-wt and M9-fps90-wt strains, which were sensitive to fPS-2 but not fPS-90, although with 10^4^-fold reduced EOP (Appendix A). The mutants M1-fps2-wt, M2-fps2-wt, and M3-fps2-wt were resistant to both fPS-2 and fPS-90. However, all the fPS-2- and fPS-90-resistant mutants were sensitive to fPS-65 (Table 1). The comparison of the genomes of the mutants to that of the wild type strain PB1 revealed five different types of deletions that affected the *galU* gene. While an identical 12-bp deletion was present in M1-fps2-wt, M2-fps2-wt, M3-fps2-wt, M1-fps90-wt and M8-fps90-wt, a 6-bp deletion in M5-fps90-wt and M6-fps90-wt, and a 14-bp deletion in M3-fps90-wt, the whole *galU* gene was deleted in M7-fps90-wt and M10-fps90-wt, that had 7.9-kb and 4.3-kb deletions over the *galU* locus, respectively (Table 1). While in M4-fps90-wt, an insertion of a T in the middle of the *ompF* gene caused a frame shift mutation, in M9-fps90-wt, a single nucleotide substitution (T to C) was within the *ompF* promoter region (Table 1). To exclude the possibility that a second-site mutation had caused the resistance phenotypes, five of the *galU*-mutants (M1-fps90-wt, M3-fps90-wt, M5-fps90-wt, M7-fps90-wt, and M10-fps90-wt) and both *ompF*-mutants (M4-fps90-wt and M9-fps90-wt) were chosen for *in trans* complementation experiments. The sensitivity of all the *galU*-mutant strains to fPS-2 and fPS-90 was restored when they were complemented with plasmid pTM100-galU [33] containing the intact *galU* gene of *Y. enterocolitica* serotype O:3 (Appendix A). Additionally, M4-fps90-wt and M9-fps90-wt retained the sensitivity to fPS-90 when complemented with the plasmid pTM100-ompF carrying the intact *ompF* gene of *Y. pseudotuberculosis* strain PB1 (Appendix A). These results indicated that the GalU activity is required to maintain the receptor structure of both fPS-2 and fPS-90 phages. GalU is UDP-glucose pyrophosphorylase responsible for the synthesis of UDP-glucose that is one of the precursors required for LPS core oligosaccharide biosynthesis [34]. Thus, the *galU* mutants have a deep rough LPS phenotype, devoid of O-antigen and with a truncated LPS inner core oligosaccharide composed of lipid A, KDO/KO-residues (octulosonic acids), and two heptoses [34]. The results demonstrated that also the OmpF protein serves as a receptor for both fPS-90 and fPS-2. OmpF is a major outer membrane protein that forms a trimer with a central channel through which the bacteria import nutrients, and export waste products and some antibiotics [35].

In general, many studies have reported that most T-even phages walk on the bacterial surface and the tips of the long tail fibres (LTFs) bind reversibly to a suitable outer membrane protein (OMP). This step, which brings the phage closer to its host cell surface, is followed by an irreversible attachment of the short tail fibres (STFs) to LPS [36]. Among OMPs, OmpC, along with LPS, are well-known as the receptors of T4-like phages [36,37,38], and OmpF is found to be recognized by T2-like phages as a receptor [7,39].

As isolation of spontaneous phage-resistant mutants against fPS-65 with the ordinary plating method was not successful, we determined the growth curves of *Y. pseudotuberculosis* PB1 bacteria in the presence of fPS-65 using the Bioscreen system. We reasoned that after the initial bacterial lysis, at later time points spontaneous phage-resistant mutant could start growing in the cultures. When visualizing the growth curves (Figure 5), from the initial OD_600_ of 0.2, the control bacteria reached stationary phase after 20–25 h of incubation. We noticed that after the initial lysis phase of the bacteria that took place after 10–15 h incubation, the growth curves of the phage-infected cultures reached a minimum after 20–25 h, followed by new growth phase until 40 h, an indication of the appearance of spontaneous phage-resistant mutants. As described in Section 2.9, a 2 mL co-culture of *Y. pseudotuberculosis* PB1 and fPS-65 with an approximate MOI of 0.001 was set up and after 48 h samples were streaked on LA plates to recover surviving bacteria. After 6 days, altogether 12 colonies growing on LA-plates were streaked on CIN agar plates, five of which failed to re-grow. Among the remaining clones only three strains (M1-fps65-wt, M2-fps65-wt, and M3-fps65-wt) were resistant to fPS-65. Whole genome sequencing analysis revealed that both M1-fps65-wt and M3-fps65-wt had an identical out-of-frame deletion in the *hldE* gene. HldE is the bifunctional D,D-heptose 7-phosphate kinase/D,D-heptose 1-phosphate adenylyltransferase [40]. In M2-fps65-wt there was a deletion in the *hldD* gene that encodes for the ADP-glycero-*manno*-heptose 6-epimerase [40]. Thus, both the *hldE* and *hldD* gene products are involved in the biosynthesis of ADP-L-β-D-heptose, the substrate for the proximal sugar residue of the LPS inner core. Thus, mutations in either of these genes would yield the same deep rough heptoseless LPS phenotype, i.e., with lipid A substituted just with the octulosonic acid residues [40,41,42]. These results suggested that the heptose region of the LPS inner core functions as receptor for fPS-65. The *hldE* mutants were fully complemented *in trans* by a plasmid pTM100-hldE that carries a fully functional *hldE* gene of *Y. enterocolitica* serotype O:3 strain 6471/76 [33]. Of note, LPS functions as a virulence factor in *Y. pseudotuberculosis* [43], suggesting that any phage-resistant mutant would have lost virulence that might be important to keep in mind in the case of using these phages for phage therapy.

### 3.7. Heat-Stability of the fPS-65 Receptor

Heat treatment of bacteria can be used to differentiate whether the phage receptor is a heat-sensitive protein or heat-resistant carbohydrate structure such as LPS. To verify that the LPS inner core is the receptor of fPS-65 we compared the ability of boiled and non-boiled bacteria to adsorb the phages (Figure 6). The results demonstrated that heat-treatment had no influence on the adsorption of the phages on bacteria, indicating as expected that the phage receptor is heat-resistant. This supported the sequencing results.

### 3.8. The Receptor-Binding Proteins of the Phages

Generally, T-even phages encode Gp37 and Gp38 as the main components of the LTFs that are critical for host attachment [44,45,46]. In phage T4, Gp38 acts as a chaperon that facilitates the trimerization of Gp37, while in T2-like phages, Gp38, after its function as a chaperone, will remain attached as the most distal part of the fibre structure. Thus in these phages, Gp38 is the adhesin that mediates the phage-host interaction [47,48]. While fPS-2 and fPS-90 both use OmpF and LPS core as receptors but had differences in their host range sensitivity data, the fPS-65 receptor is the heptose region of LPS inner core. To find out whether the receptor specificities of fPS-2, fPS-90, and fPS-65 could be explained by the predicted amino acid sequences of their respective LTF components, the identified Gp37 and Gp38 homologs of the fPS-phages were aligned for comparison. The alignment of the Gp37 homologs (Gp245, Gp246, and GP248 of fPS-2, fPS-65, and fPS-90, respectively) showed that the Gp245 amino acid sequence of fPS-2 is quite different from the Gp246 of fPS-65 and Gp248 of fPS-90, while the two latter are more similar to each other. Thus, this did not reflect at all the receptor specificities of the phages. Overall, the 60 most N-terminal and the 120 most C-terminal amino acid residues of the Gp37 homologs of the fPS phages were highly identical, and the sequences in between were more different (Figure 7a).

In contrast, the host receptor specificity was better reflected in the multiple sequence alignment of the Gp38 homologs. When carrying out HHpred [49] search with Gp246 of fPS-2, the best hit was the Gp38 homolog (DOI: 10.2210/pdb6F45/pdb) of *Salmonella* phage vB_SenM_S16 (S16 for short) [48]. Multiple sequence alignment of the Gp38 fPS-homologs with that of S16 is shown in Figure 7b. The Gp38 homologs of the T2-like phages, including S16, have a modular organization: a conserved N-terminal domain forms a highly hydrophobic surface that binds to the similarly highly hydrophobic Gp37 trimeric tip of the tail fibre. The C-terminal domain in Gp38 homologs that determines the host specificity has a segmented structure [48]. The C-terminal domain contains a series of ten conserved polyglycine motifs that flank the five distal hypervariable loops that are centrally involved in the receptor-binding specificity of the T2-like phages [45,48].

The alignment in Figure 7b demonstrates that the sequence similarity of the N-terminal attachment domain of the fPS phage Gp38 homologs is high including the conserved tryptophan residues that insert like prongs into hydrophobic pockets on the gp37 distal fragment [48]. Thus, it is likely to expect that the Gp38 docking site of the Gp37 homologs should share structural features. In S16, after the trimerization of the Gp37 homolog, the last 113 residues comprising the intramolecular chaperone (IMC) is autocleaved and this exposes the Gp38 docking surface to the distal end of Gp37 structure. Alignment of the Gp37 homologs (Figure 7a) reveals that the sequences preceding the IMC sequence are very different in fPS-homologs when compared to that of S16, with the exception of the five last residues. The cleavage site in S16 is between these two serines in the sequence **DLNVS^SDRRIKK**. This sequence aligns to the **DVYIR^SDGRLKI** sequence in the fPS-phages and the cleavage site must thus be between the arginine and serine residues. The DVYIR sequence is predicted to form a β-strand and that the three copies of it in the trimer will form to which the gp38 N-terminal attachment domain will dock. Therefore, we predict that the fPS Gp37 homologs are cleaved after the DVYIR sequence. The C-terminal part of the sequence forms the IMC that dissociates and opens a binding site for the Gp38 homolog [48,50]. The DVYIR residues will form a flat triangular tip with indentations for binding of the conserved tryptophans of the gp38 adhesins. Interestingly the preceding structures of the Gp37 homologs in fPS-phages are novel indicating a very rare occurrence of receptor-binding protein gene mosaicism where the “cut” goes through a beta-strand that precedes the IMC and adhesin modules. Such a tight cut is not common, usually when these modules are exchanged between organisms, a much larger part of the beta-helix is captured.

Comparison of the Gp38 homolog C-terminal sequences reveals extensive sequence variability in the fPS-65 sequence in the predicted distal loops 1–5 (Figure 7b). In contrast, the sequences of phages fPS-2 and fPS-90 are identical except for a single Q -> A change after the loop 5. This single amino acid difference could explain the slightly different host ranges of fPS-2 and fPS-90. The highly variable loops in fPS-65 explain perfectly its unique receptor and host range.

## 4. Conclusions

In this work, we characterized three *Y. pseudotuberculosis*-infecting phages, fPS-2, fPS-65, and fPS-90, originally isolated from pig stool samples collected from Finnish pig farms [8]. The genomic sequences of the phages were 85–92% identical to each other. Based on the electron microscopy and the phylogeny analysis, the phages can be classified as new members in the genus Tequatroviruses within the *Myoviridae* family under the *Caudovirales* order. The phages fPS-2, fPS-65, and fPS-90 lysed 21, 33, and 18 of the 56 *Y. pseudotuberculosis* strains tested, respectively, and, in addition, one *E. coli* strain indicating that the phages have a relatively broad host range. On the genomic level, the genomes of the phages showed a high level of similarity between them and to T4 phage. While OmpF and LPS outer core oligosaccharide were identified as receptors on the bacterial surface for both fPS-2 and fPS-90, the fPS-65 receptor structure required the presence of the heptoses of the LPS inner core. These receptor specificities were reflected to the amino acid sequence differences in the Gp38 homologs of the fPS-phages.

In general, being lytic and void of any genes encoding for toxicity, antibiotic resistance, or lysogeny, these phages can be used safely in phage therapy. Furthermore, as fPS-65 is also able to infect *Y. pestis*, it could be considered for diagnostic and therapeutic purposes against the notorious pathogen, the etiologic agent of the Black Death.

## Figures and Tables

**Figure 1 viruses-13-00296-f001:**
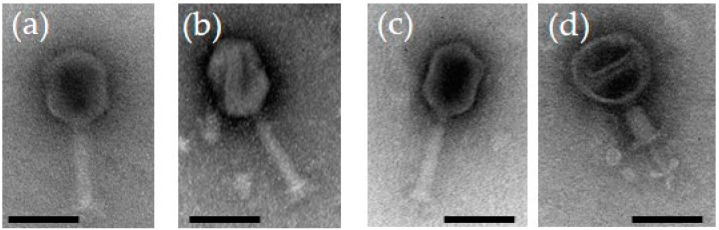
Transmission electron micrographs of negatively stained bacteriophages: (**a**) fPS-2; (**b**) fPS-90; (**c**) fPS-65; (**d**) fPS-65 with a contracted tail. The scale bar is 100 nm.

**Figure 2 viruses-13-00296-f002:**
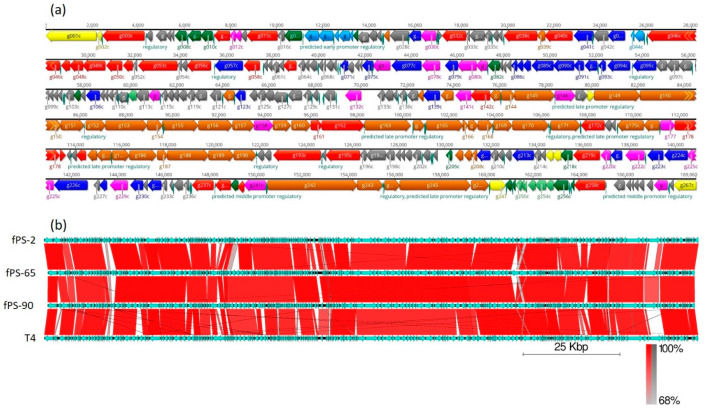
Gene organization in the fPS-phage genomes. Panel (**a**) Detailed genomic map of phage fPS-2. The gene colors indicate predicted functions of their products: *Red*, RNA and DNA maintenance proteins; Blue, dNTP/NTP metabolism associated proteins; Light blue, toxin-antitoxin system and phage immunity; Brown, phage particle-associated proteins (PPAPs); Yellow, membrane/peptidoglycan active proteins; Pink, endonucleases, homing endonucleases and exonucleases; Dark green, regulatory proteins; Light green, anti-restriction proteins; Grey, hypothetical proteins or phage proteins. The figure was generated using Geneious 10.2.6. Panel (**b**). BLAST alignment of the fPS-genomes with that of phage T4 drawn using the EasyFig tool [26]. The annotated genes and their orientations are indicated by filled arrows and the sequence BLAST identities of the corresponding genes between the phages by the red-shades quadrangles.

**Figure 3 viruses-13-00296-f003:**
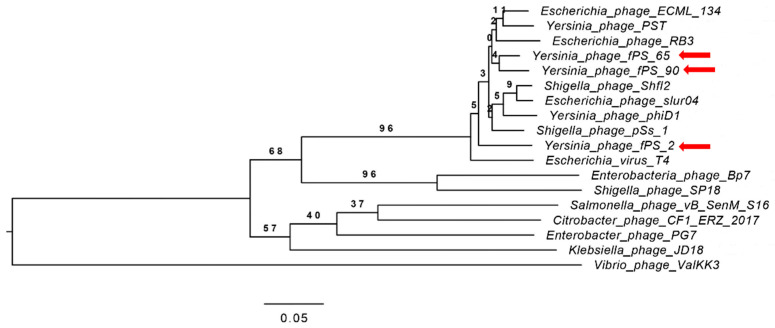
Phylogenetic tree generated by VICTOR using the complete genome sequences of fPS-phages and those of 15 other phages from the *Myoviridae* family. Phages fPS-2, fPS-65 and fPS-90 are indicated with red arrows.

**Figure 4 viruses-13-00296-f004:**
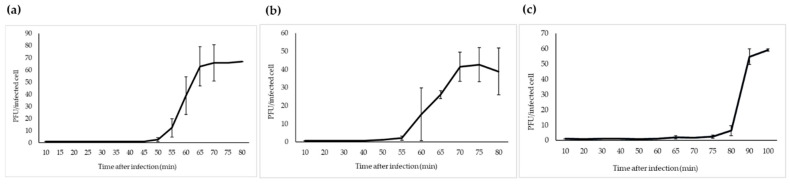
One-step growth curves of fPS-phages in *Y. pseudotuberculosis* O.1a strain PB1: (**a**) fPS-2; (**b**) fPS-65; (**c**) fPS-90. Error bars represent standard deviations (SD).

**Figure 5 viruses-13-00296-f005:**
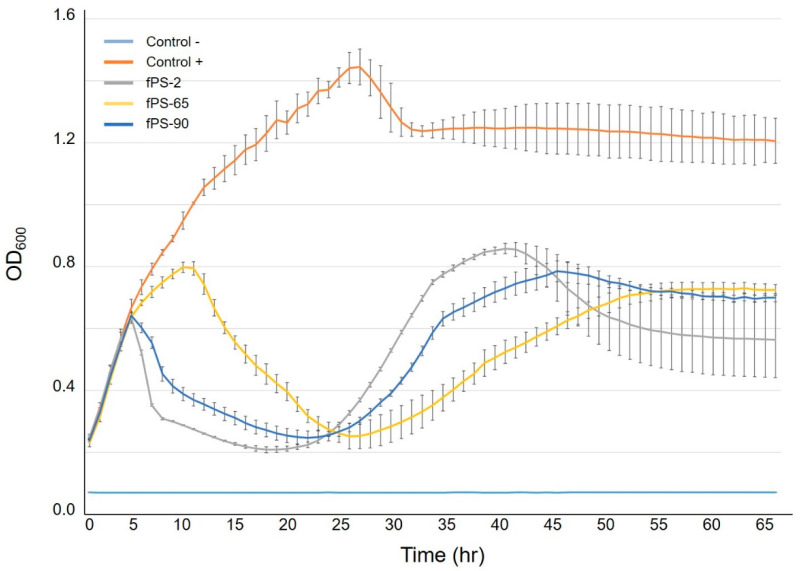
Growth curves of *Y. pseudotuberculosis* O:1a strain PB1 infected with phages fPS-2, fPS-65 and fPS-90 at a MOI of 0.001. Each curve represents the average results for five replicates. Error bars represent standard deviation (SD).

**Figure 6 viruses-13-00296-f006:**
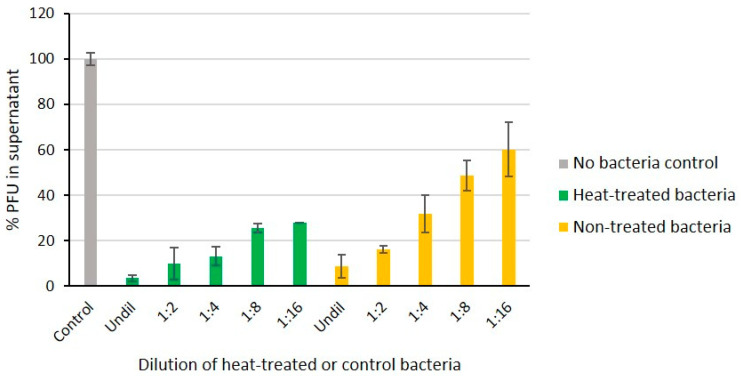
The fPS-65 receptor is heat-stable. Phages were incubated 15 min with different dilutions of heat-treated and non-treated bacteria. Shown are percentages of free phages in the supernatants after centrifugation when compared to no-bacteria control set to 100%. The boiled bacteria adsorbed the phages as well as or better than viable bacteria indicating that the phage receptor is intact. Error bars represent the standard deviations for the average plaque numbers in two parallels of two independent experiments.

**Figure 7 viruses-13-00296-f007:**
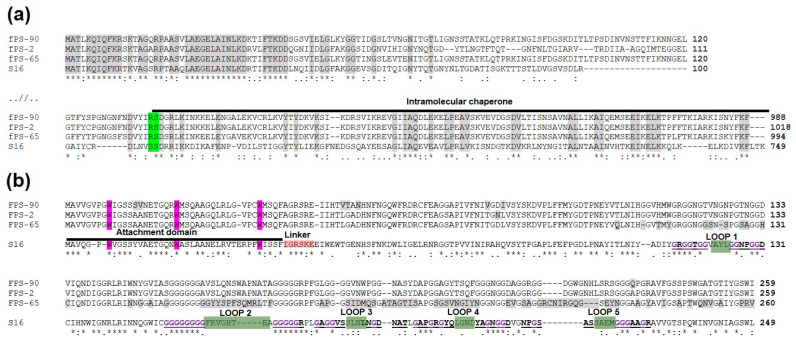
Multiple sequence alignments of Gp37 and Gp38 homologs of phages fPS-2, fPS-65, fPS-90 and Salmonella phage S16. Below the alignments symbols are used to indicate the similarity of the aligned amino acids: “*” indicates perfect alignment; “:” indicates that aligned amino acids belong to a group exhibiting strong similarity; “.” indicates that aligned amino acids belong to a group exhibiting weak similarity. Panel (**a**). Multiple sequence alignment of the N- and C-terminal sequences of the Gp37 homologs. Identical amino acids in all four sequences are highlighted in grey. The autocleavage site is highlighted in green and the intramolecular chaperone indicated by black line. Panel (**b**). Multiple sequence alignment of the sequences of the Gp38 homologs. Different functional regions are indicated for the S16 protein. The N-terminal attachment domain is indicated by a black line, and its critical N-terminal tryptophan residues are highlighted in purple. The linker sequence is highlighted in pink. The predicted receptor-binding loops 1–5 are indicated by green highlighting, and the flanking polyglycines in bold purple color with underlining. The differences between the fPS homologs are highlighted in grey.

**Table 1 viruses-13-00296-t001:** Overview of phage-resistant mutants and their sensitivity to the fPS-phages.

Mutant Strain	Storage#	Gene	Mutation, Reference to Genome of PB1 (acc no NC_010634)	Sensitivity to		
				fPS-2	fPS-65	fPS-90
**M1-fps2-wt**	6582	*galU*	del-TAAAAGAGATCA, 1,869,879	Res *	Sens **	Res
**M2-fps2-wt**	6583	*galU*	del-TAAAAGAGATCA, 1,869,879	Res	Sens	Res
**M3-fps2-wt**	6584	*galU*	del-TAAAAGAGATCA, 1,869,879	Res	Sens	Res
**M1-fps90-wt**	6585	*galU*	del-TAAAAGAGATCA, 1,869,879	Res	Sens	Res
**M3-fps90-wt**	6586	*galU*	del-AAGAGATCAAAAAT, 1,869,882	Res	Sens	Res
**M4-fps90-wt**	6587	*ompF*	C → CT, 1,721,035	EOP ***	Sens	Res
**M5-fps90-wt**	6650	*galU*	del-AAAGAG, 1,869,881	Res	Sens	Res
**M6-fps90-wt**	6651	*galU*	del-AAAGAG, 1,869,881	Res	Sens	Res
**M7-fps90-wt**	6652	*galU*	ca. 7.9 kb del (1,870,248..1,878,146)	Res	Sens	Res
**M8-fps90-wt**	6653	*galU*	del-TAAAAGAGATCA, 1,869,879	Res	Sens	Res
**M9-fps90-wt**	6654	*ompF*	promoter region, T → C, 1,721,588	EOP	Sens	Res
**M10-fps90-wt**	6655	*galU*	ca. 4.3 kb del (1,865,646..1,869,969)	Res	Sens	Res
**M1-fps65-wt**	6941	*hldE*	147 bp del (3,957,759..3,957,905)	Res	Res	Res
**M2-fps65-wt**	6942	*hldD*	517 bp del (61,956..62,473)	Res	Res	Res
**M3-fps65-wt**	6943	*hldE*	147 bp del (3,957,759..3,957,905)	Res	Res	Res

* Res: resistant; ** Sens: sensitive; *** EOP: 10^4^-fold reduced EOP.

## Data Availability

Not applicable.

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
