# Peer review of "T4-like Bacteriophages Isolated from Pig Stools Infect Yersinia pseudotuberculosis and Yersinia pestis Using LPS and OmpF as Receptors"

_viruses, 2021, doi:10.3390/v13020296_

Round 1

Reviewer 1 Report

The paper presents an interesting investigation of some T4 like viruses and

emerging mutants. The deterministic development of mutants presumably 

in Fig. 5 would be nice to supplement with numbers on how big the samples was (i.e. how many bacteria at eg. OD=0.1), optimally with some estimate of mutation frequency. Look like the red curve (fps65) represent mutants that are less likely to evolve. But overall fig. 4 and 5 (and table 1) presents nice results that would be of interest to the community.

I recommend publication.

Author Response

See to attached document

Reviewer 2 Report

The authors present a well written manuscript describing the characterization of three Yersinia pseudotuberculosis phages. Sequencing of phage resistant mutants show that mutations and deletions in genes involved in LPS metabolism influence the phage infection. 

Remarks: 

Section 2.6 One-Step growth: Please provide the multiplicity of infection (MOI) used during this assay. 

Section 2.7: Please provide the MOI used during this assay. What was the bacterial concentration used?  

Section 2.9: How was the resistance of the bacteria tested to the phage? What bacterial concentration was used to perform the liquid infection assay to obtain phage resistant mutants? How was the comparative genomics analysis performed of the mutants compared to the ancestral bacterium? 

Section 3.1 Electron microscopy: Please edit the figures so tat the (a), (b), (c) are included in the image itself. Please provide a discussion. Suggestion: merge section 3.1 and 3.2. 

Section 3.2:

The authors describe that the investigated phages contain linear, circularly permuted DNA, but there is no experimental data to support this claim.  

Please provide a figure that shows a BLAST comparison of the phages isolated in this project and a comparison to the  closest relative phage described in literature. Highlight in this figure the five modules described (line 206). This gives an overview of the homology between the three phages as well as to a reference phage. 

Section 3.4: 

Suggestion: extend the discussion to give more insights in the host range of other T4 phages. 

Section 3.5: 

Create one graph showing the one-step growth curves of all phages in different colors. This will aid in understanding the differences in growth characteristics between the different phages. Show full error bars (both positive and negative). Write a discussion that compared the growth kinetics of the phages described in this research and other, highly similar, phages in literature. 

Section 3.6: 

line 286 (and line 331): The authors describe that OmpF serves as a receptor for both fPS90 and fPS2. However, Table 1 shows that an alteration in the gene sequence of ompF does not influence the infectivity of fPS2. Moreover, this seems to be the only difference between phages fPS2 and fPS90. Could you please further elaborate on this? Please include a discussion of the function of Omp's in the infectivity of T4 and T4-like phages. 

line 280: Show figures of the resistance tests performed with the mutants e.g. drop tests and figures of the complementation assay. Currently there are no data presented that support the claims as presented in this section. 

Figure 5: please provide the MOIs used in this assay instead of the phage dilutions. 

Section 3.7 figure 6:

Please show the full error bars. Suggestion: instead of using the number of free phage, show the percentage of free phage. As only 2 repeats have been performed, an additional repeat would strengthen the results presented in this assay. 

Conclusion: 

line 389: please change "many" to a correct figure. 

Author Response

See the attached document

Reviewer 3 Report

The authors investigated three T4-like phages infecting Yersinia. It's interesting these phages exhibited broad host ranges thus they may open a new window for phage-based microbial control. It is reasonable that the research focuses on the receptors with appropriate approaches. 

My major comments are as follows,

First, in the introduction, the authors should do more literature review on T4-like phages and potential Yersinia phages. Especially, the receptors of these reported phages. Second, I am curious about the adsorption rate constants of these phages at different pH, temperature and salinity. Also I am curious how the affinity changes if the host is treated with proteinase and LPS depolymerase separately.  Third, the authors obtained several phage resistant mutants. I recommend the authors also compare the growth rate and biofilm formation capability between the wildtype and mutants. Such information is important for applications. 

Author Response

See the attached document

Round 2

Reviewer 2 Report

I thank the authors for considering the remarks made in the previous reviewing round. There are some minor typo's left in the manuscript: line 207-208 and line 327. 

As an extra suggestion, the genome map of the phage would greatly improve if you would incorporate all genomes of the different phages along with T4, including the blast-comparison of the genomes. You can make such figures using easyfig (https://mjsull.github.io/Easyfig/)

Author Response

Thank you for the comments. Based on your suggestion we have revised figure 2 and included there the requested BLAST comparison of the genomes using the recommended EasyFig tool. We have also corrected the typos although we did not find any apparent typos in lines 207-208 but that part was also revised